# Nutraceuticals and the Network of Obesity Modulators

**DOI:** 10.3390/nu14235099

**Published:** 2022-12-01

**Authors:** Sergio Ammendola, Anna Scotto d’Abusco

**Affiliations:** 1Ambiotec di Sergio Ammendola, 04012 Cisterna di Latina (LT), Italy; 2Department of Biochemical Sciences, Sapienza University of Rome, 00185 Rome, Italy

**Keywords:** herbal nutrients, obesity, chronic inflammation, adipogenesis, lipogenesis, cell crosstalk

## Abstract

Obesity is considered an increasingly widespread disease in the world population, regardless of age and gender. Genetic but also lifestyle-dependent causes have been identified. Nutrition and physical exercise play an important role, especially in non-genetic obesity. In a three-compartment model, the body is divided into fat mass, fat-free mass and water, and obesity can be considered a condition in which the percentage of total fat mass is in excess. People with a high BMI index or overweight use self-medications, such as food supplements or teas, with the aim to prevent or treat their problem. Unfortunately, there are several obesity modulators that act both on the pathways that promote adipogenesis and those that inhibit lipolysis. Moreover, these pathways involve different tissues and organs, so it is very difficult to identify anti-obesity substances. A network of factors and cells contributes to the accumulation of fat in completely different body districts. The identification of natural anti-obesity agents should consider this network, which we would like to call “obesosome”. The nutrigenomic, nutrigenetic and epigenetic contribute to making the identification of active compounds very difficult. This narrative review aims to highlight nutraceuticals that, in vitro or in vivo, showed an anti-obesity activity or were found to be useful in the control of dysfunctions which are secondary to obesity. The results suggest that it is not possible to use a single compound to treat obesity, but that the studies have to be addressed towards the identification of mixtures of nutraceuticals.

## 1. Introduction

Obesity is an individual condition in which the percentage of fat mass exceeds the lean mass, causing the development of secondary diseases such as diabetes, kidney insufficiency, osteoarthritis and cardiovascular diseases. The body mass distribution can be ascertained by performing a bioimpedentiometry, which gathers information on the nutritional status of an individual. The three-compartment model is widely used to estimate body composition. According to this model the total body mass is divided into fat mass (FM), fat-free mass (FFM) and total body water (TWB). The relative percentage of each compartment correlates with a range that is typical of each ethnic group [1]. The differences between males and females, of same age, provide the reference intervals of Body Mass Index (BMI, kg/m^2^); generally, a BMI > 30 kg/m^2^ is considered obesity. In a sample of Caucasian people (*n* = 5635) it was observed that from young to elderly age categories, FM progressively rose by an average of 55% in males and 62% in females [2]. For Caucasians, the average fat mass does not exceed 15% in males and 27% in women. Part of the fat mass is essential, such as that which gives elasticity to cell membranes, to protect and nourish neuronal cells, and to regulate homeothermy. Another part of fat, that accumulated in adipocytes, cells specialized in storing lipids, is a source of energy. When the energy intake from food sources is higher than energy expenditure, the surplus is converted in fatty acids and stored in these cells that will proliferate, or expand their volume, in order to store more energy. The control of the balance between energy intake and expenditure is influenced by numerous factors that can cause obesity, such as those promoting the maturation of pre-adipocytes (adipogenesis) or those stimulating the lipid degradation with energy release (lipolysis). The regulation of energy expenditure, lipolysis and adipogenesis, are three of the main events whose dysregulation determines the onset of obesity [3]. Moreover, adipose tissue is composed of various cell types including adipocytes, fibroblasts, endothelial cells, and various immune cells, which are in communication each other through a paracrine and endocrine system. Preadipocytes can differentiate into white (WA), brown (BrA) and beige types (BeA) and each one has different biological activity. Moreover, the adult white cells can differentiate into brown/white-like cells, through the mechanism called transdifferentiation (browning of WA tissue) [4]. The brown and beige adipocytes have a high number of mitochondria compared to white cells and are committed to burning fat. Fibroblast and endothelial cells, surrounding the blood vessels, can cause inflammation and in turn obesity [5]. Similarly, the immune cells infiltrating inflamed adipose tissue, such as the adipose tissue macrophages (ATMs), play a causal role in obesity-induced insulin resistance. Moreover, natural killer T cells (NKs) were reported as regulators of adipose tissue inflammation in obesity. Several recruited cells contribute to increased fat mass, and it has been shown that they are associated with insulin resistance and are linked to changes in the extracellular matrix [6]. Furthermore, adipocytes and macrophages produce cathecolamines, which are at the basis of the interaction between these tissues and affect the centres of satiation [7,8]. The adipocytes secrete small molecules and proteins that regulate inter- and intra-cellular signalling. Some hormones activate lipases and induce lipolysis, while others can inhibit or promote lipogenesis. Many endogen mediators contribute to modulate these signalling elements; moreover, other factors, including the composition of the microbiota, can cause obesity. Furthermore, some intestinal microorganisms are able to promote the synthesis of short-chain fatty acids that act on the centres of hunger and affect the energy balance. Unlike an orchestra, there is no single conductor but many mediators, which can determine the hypertrophy of adipocytes or promote hyperplasia.

Nutraceuticals are natural compounds, generally having a molecular size not exceeding 3000 Da. They are not involved in primary metabolism and exhibit a health-promoting bioactivity. Computational chemistry predicts about 400,000 bioactive compounds, belonging to several chemical families, of pharmaceutical interest [9]. The complexity of obesity involves the coexistence of different signals including the epigenetic changes [10]. Therefore, an ideal composition of nutraceuticals should be able to act on all possible pathways that affect the increase in non-essential fat mass. To find a match between targets and nutraceuticals is very challenging and such studies are often based on ethnopharmacology or a cyclic process of chemical screening. In the present narrative review, we describe the effects of food supplements and teas according to popular traditions and nutraceutical composition obtained after in vitro or in vivo studies. We describe the natural products, especially herbs, containing nutraceuticals that have been shown to be helpful in preventing or treating non-genetic obesity and their putative mechanism of action. Some phytocomplexes act on lipolysis but do not prevent the hyperplasia of adipocytes. Other compounds induce thermogenesis but do not prevent the hypertrophy of adipocytes. Others are prebiotics that promote the growth of fat-consuming microorganisms, but they do not prevent the stem cells from becoming adipocytes [11]. Moreover, extracts obtained from different parts of the plants act through very different mechanisms, even if all of them share similar anti-obesity effects.

We never found the description of a nutraceutical or a composition thereof working efficiently after two years of administration. Furthermore, the compounds acting on the same target have very distant chemical structures. We grouped these biomolecules according to the mechanisms of action or similar effects on candidate targets of obesity. Although some studies have been conducted in vitro or in animal models, the results are useful in establishing a method for composing an anti-obesity agent. The ideal composition should inhibit adipogenesis and promote thermogenesis, activate lipolysis and reduce appetite. Furthermore, after a long period of administration, a composition of cooperating nutraceuticals should not induce tolerance and adaptation to agents and should have acceptable side effects.

## 2. Results

Adipose tissue (AT) is composed not only of adipocytes but also of macrophages, endothelial cells and preadipocytes [12]. Mesenchymal stem cells (MSCs) are the major source of adipocytes, and they initially differentiate into preadipocytes that are committed to the adipogenic lineage. Consequently, preadipocytes give rise to enlarged mature adipocytes that can synthesize lipid droplets, secrete specific adipocyte factors, and regulate energy metabolism. The differentiation of adipocytes from MSCs is believed to play a vital role in maintaining the adipose tissue homeostasis. The endothelial cells, which line blood vessels, are stimulated by adipokines and can cause unhealthy fat tissue expansion by crosstalking with adipocytes [13]. Macrophages resident in the adipose tissue are distinguished by two subtypes, the M1-like and M2-like macrophages. The M1 macrophages produce pro-inflammatory cytokines, such as Tumor Necrosis Factor (TNF)-α, interleukin IL-6 and MCP-1, which contribute to the development of insulin resistance. On the other hand, M2 macrophages are anti-inflammatory, are involved in the maintenance of tissue homeostasis, and are typically present in the adipose tissue of non-obese individuals. The imbalance between M1 and M2 macrophages has been found to be responsible for chronic inflammatory milieu in adipose tissue and insulin sensitivity. In lean individuals, macrophages dispersed throughout adipose tissues are predominantly polarized toward the M2 phenotype, while in obese individuals, an elevated number of infiltrating M1 subtype macrophages are present [14]. Different mechanisms control adipocyte differentiation and macrophage differentiation and infiltration. AMPK and PPAR pathways and the decrease of inflammatory response are the main targets of the anti-obesity strategies (Figure 1).

### 2.1. The Activation of AMPK Pathway

The AMP-activated protein kinase (AMPK) pathway is a crucial energy sensor that regulates energy metabolism in multiple tissues. AMPK is a conserved, ubiquitously expressed, heterotrimeric serine/threonine kinase whose short-term activation has multiple beneficial metabolic effects. Different subunits take part in the complex, which plays critical roles in regulating growth and reprogramming the metabolism [15].

The alpha subunit can phosphorylate itself in the Thr172 position or self-inhibit. The beta subunit has a glycogen-binding domain (GBD) and a domain that is able to modify its conformation by favoring the binding of the alpha subunit to the gamma. In the gamma subunit there are four AMP/ATP allosteric binding sites that regulate the activity of the complex itself. When the intracellular energy decreases, the AMP level is very high compared to ATP and ADP levels. In this condition, the AMP binds a cystathionine-beta-synthase (CBS) domain present in the gamma subunit, inducing a change in its conformation which in turn promotes the self-phosphorylation of the alpha subunits [16]. Active AMPK binds other kinases, such as calcium/calmodulin-dependent protein kinases (CaMKs) and Transformation Growth Factor (TGF)-β activated kinase-1 (TAK1).

Furthermore, AMPK is involved in the cellular processes including autophagy and cell polarization [15]. It is activated when modest decreases in ATP production result in relative increases in AMP or ADP concentration. In response, AMPK promotes catabolic pathways to generate more ATP to inhibit anabolic pathways. In brown adipocytes the activation of AMPK signaling stimulates thermogenesis. This mechanism corresponds to fat burning, which produces an increase in temperature, instead of ATP, through thermogenin or an uncoupling protein1 (UCP1) and, in turn, induces the increase in energy expenditure, leading to weight loss. However, it should be taken into consideration that the energy consumption, due to thermogenesis, represents only 10% of total energy expenditure. Furthermore, the subunit AMPKα appears to regulate the browning of white adipocytes, increasing the energy expenditure [17,18]. Activation of AMPK has been proposed as an attractive strategy for the treatment of obesity and its complications. However, some careful consideration must be taken since, in an animal model, the chronic and non-specific tissue activation of the AMPKγ2 subunit resulted in inducing hyperautophagy and dysglycemia [19]. Some nutraceuticals present in herbal extracts resulted in an ability to activate the AMPK pathway. The characterization of active substances in extracts from anti-obesity plants revealed that they reduce the absorption of glucose and, at the same time, increase its energy consumption by acting on the AMPK pathway. The *Gelidium* species, a red alga, has been found to act on the AMPK-PR domain-containing the 16-uncoupling protein-1 pathway (PRDM16-UCP1), with a healthy effect in obese people. Moreover, in a mice model of obesity, an extract of the red alga was found to enhance the effect of Orlistat, a known lipase inhibitor [20,21]. An extract from *Paullinia cupana* seeds has been found to reduce obesity by inducing the overexpression of the UCP-1 via activation of AMPK, too [18,22]. Thermogenesis can also be induced when white adipocytes differentiate in beige via a browning mechanism. Polyphenols from *Zingiber officinale* have been found to be able to promote adipocyte browning in an animal model [23]. WAT browning is regulated by several genes, such as UCP1, PR domain containing 16 (PRDM16), cell death activator CIDE-A and PGC-1a, whose expression is under the control of the AMPK and sirtuin 1 (SIRT1) protein [24].

*Panax ginseng* has been shown to contain the Ginsenoside Rg1, which is able to promote browning, by inducing UCP1 expression and by modulating the transcription factor PRDM16 and PGC1α [25,26]. However, this modulation depends on Irisin, an adipomyokine, which promotes WAT browning, acting both in adipose and muscle tissue [27,28]. Other plants have been found to induce heat dissipation, and among them the root extract from *Pueraria montana* var. lobata has been shown to stimulate thermogenesis in BrAs; however, other species of *Puerariae* have also been described as having a similar activity [29,30].

Many natural compounds, such as Capsaicin, Curcumin, polyphenols found in cranberry and flavonoids in rose, have been proposed as adipocyte browning agents [31]. Mostly used are nutraceutical compositions of extracts, such as a composition of radix from *Angelicae sinensis* and rhizome from *Zingiber officinale*. *Z. officinale* has also been widely reported to activate the AMPK signalling pathway [32,33]. A similar effect was observed using an ethanolic extract of *Scutellaria baicalensis;* in this case, however, the biological effect relates to the wingless integrated protein (WNT)/β-catenin pathway. Baicalin, and Baicalein, have been found to be active against metabolic syndrome by activation and upregulation of AMPK and PPARγ [34,35]. The AMPK pathway is also stimulated by *Iris rossii*, which inhibits lipid accumulation; its extract, containing mangiferin, was also found to inhibit preadipocyte differentiation [36,37]. *Salvia hispanica*, also known as chia, extracts act on glucose metabolism by increasing AMPK mRNA [38,39].

However, the AMPK subunits are differentially and tissue-specifically expressed, and AMPK activators (agonists) fail to function over a long time due to their non-specific activity [40] (Table 1).

Moreover, the AMPK pathway is related to the fat mass and obesity-associated (FTO) gene, which regulates the lipid accumulation in skeletal muscle via a N^6^-Methyladenine demethylation mechanism [41]. This type of demethylation is dependent on epigenetic changes [42]. However, the FTO gene has been also associated with genetic polymorphisms that affect the individual response to anti-obesity drugs [43].

An extract from *Angelica sinensis* and two extracts from *Solanum melogena* and *S. aethipicum*, respectively, have been shown to contribute to reducing obesity via modulation of the FTO gene [44,45,46,47] (Table 2).

### 2.2. The Inhibition of the PPAR Pathway

The peroxisome proliferator-activated receptors (PPARs) are transcription factors, grouped within the ligand-inducible nuclear hormone receptor superfamily, that positively and negatively regulate gene expression in response to the binding of a diverse array of lipid-derived hormones and metabolites. Three major isoforms, PPARα (or NR1C1), PPARβ/δ (or NR1C2), and PPARγ (or NR1C3), are implicated in the control of lipid metabolism, and they are also considered insulin sensors [48]. The different PPAR subunits are expressed in many tissues, including adipose, liver, muscle, and endothelial tissues. PPARα mainly influences fatty acid metabolism, and its activation lowers lipid levels, while PPARγ is mostly involved in the regulation of adipogenesis, the energy balance, and lipid biosynthesis, and its activation affects insulin resistance [49]. PPARβ/δ participates in fatty acid oxidation, mainly in skeletal and cardiac muscles, and regulates blood glucose and cholesterol levels. PPARs can also form heterodimers with retinoid X receptor (RXR) modulating the expression of genes involved in lipid metabolism, adipogenesis, maintenance of metabolic homeostasis, and inflammation. Activated cytoplasmic PPARs can translocate into the nucleus, where they bind DNA responsive elements, either as a homodimer or heterodimer complexed with retinoid acid receptor (RXR), and each transcriptional complex needs coactivators [50,51]. Moreover, the PPAR transcriptional activity can be modulated through a nongenomic cross talk with phosphatases and kinases, including ERK1/2, p38-MAPK, PKA, PKC, AMPK, and GSK3 [48]. Moreover, the members of the PPAR family also modulate basic processes, such as proliferation, differentiation, and postnatal development [52].

Activation of PPARγ by thiazolidinediones can reduce insulin resistance and hyperglycemia in type 2 diabetes, but these drugs can also cause weight gain. Apparently, a moderate reduction of PPARγ activity, observed in heterozygous PPAR γ-deficient mice or in the Pro12Ala polymorphism in human PPARγ protein, has been shown to prevent insulin resistance and obesity induced by a high-fat (HF) diet. Furthermore, it has been observed that a moderate reduction of PPARγ with an RXR antagonist or a PPARγ antagonist decreases triglyceride (TG) content in white adipose tissue, skeletal muscle and liver. However, severe reduction of PPARγ by treatment of heterozygous PPARγ-deficient mice with a RXR antagonist or a PPAR antagonist depletes white adipose tissue and markedly decreases leptin and adiponectin levels and energy dissipation, which increases TG content in skeletal muscle and liver, thereby leading to the re-emergence of insulin resistance [53].

PPARγ is considered the master activator of adipogenesis, and its inhibition is a preferred strategy to screen anti-obesity agents. PPARγ inhibitors were shown to be useful in treating obesity, through the inhibition of expression of the adipose tissue genes, but the drugs can have undesirable effects in other tissues. PPARγ activators are used as insulin sensitizers in order to improve insulin resistance in diabetic patients. PPARγ comprises seven gene isoforms, among them PPARγ1 and PPARγ2, which are prevalently expressed in adipocytes where they are involved in lipid metabolism and cell differentiation, and in macrophages where they trigger the initial inflammatory response. PPARγ2, which is highly expressed in adipocytes, is the key fat-selective PPAR subtype, when it is phosphorylated at Ser273 promotes the lipogenesis; thus, it is a good target of the anti-obesity agents. Many studies showed that both isoforms, γ1 and γ2, are increased in obese people, whereas other studies have stated that only the γ2 isoform is increased. These inconsistent results could depend on the different gene expression regulation in different tissues and on post-translational modification of PPARγ isoforms. Ligands blocking the Ser273 phosphorylation in γ2 improve insulin resistance, whereas they induce weight gain; thus, they have been thoroughly studied [54]. Moreover, the block of Ser273 phosphorylation positively correlates with increased expression of the growth differentiation factor-3 (GDF-3), which is pro-inflammatory and promotes fat accumulation [55,56]. Several herbs have been described to act on PPAR pathway, among them the *Spermacoce hispida*, the seeds of which contain deacetylasperulosidic acid, an iridoid glycoside, acting on the PPARα and exhibiting an antihyperlipidemic activity [57]. The seeds of *S. hispida* significantly reduce the proliferation of mature adipocytes and decrease fat accumulation. A similar effect was shown by *Rosa rugosa* extract, rich in flavonoids, which reduces hypertriglyceridemia by acting as PPARα agonist [58,59]. The polymethoxyflavones (Sinensetin, nobiletin, 3,5,6,7,8,3′,4′-Heptamethoxyflavone, and Tangerine) from *Citrus reticulata*, ‘Chachi’ peels, exhibit biological effects similar to those observed for *S. hispida* and *R. Rugosa;* however, their mechanism of action is still to be elucidated [60,61]. Apparently, the *Nigella sativa* oil, containing a Thymoquinone, a PPARγ agonist, decreases the expression of TNF-α and adipokines in obese and overweight people [62,63]. *Astragali membranaceus* var. mongholicus increase thermogenesis via PPARγ activation [64,65]. Moreover, the tissue-specific expression of PPAR subunits account for healthy effects on short-time and failure on long-time and chronic use (Table 3).

Some nutraceuticals have been shown to affect the PPAR signaling via different patterns; some herbal extracts modulate the CCAAT-enhancer-binding (C/EBP) alpha and beta subunits [66]. *Ziziphus jujuba* extract, containing apigenin, betulinic acid, and maslinic acid, was linked with a decrease of protein levels of PPARγ and C/EBPα. *Ziziphus jujuba* fructus has been shown to stimulate adipocyte browning with the effect of stimulating thermogenesis [67].

The theobromine from *Theobroma cacao* beans reduces the expression of C/EBPα and in turn PPARγ, and the total effect is to stimulate thermogenesis via activating the β_3_-Adenergic receptor [68,69]. The isoimperatorin from *Angelica dahuricae* significantly increased mRNA expression, and protein production, of the C/EBPα and PPARγ [70,71]. A fraction of *Bidens pilosa* extract containing cytopiloyne has been found to reduce adipogenesis by inhibiting expression of PPARγ and C/EBP [72]. An extract, rich in alkaloids from *Coptis chinensis,* showed an anti-adipogenic activity linked to its effect to downregulate C/EBP-α and PPAR-γ factors [73]. An extract, rich in terpenoids from the *Isodon adenantha*, has been shown to inhibit adipogenesis in 3T3-L1 and mouse embryonic fibroblast models. Adenanthin, an *ent*-Kaurane diterpenoid, decreases inflammation and ROS by inhibiting the nuclear factor kappa-light-chain-enhancer of activated B cells (NF-κB) signalling [74]. NF-κB can interact with the PPARγ subunit, promoting cell inflammation [75] (Table 4).

Other herbal extracts indirectly act on the PPAR pathway through very different biological targets. An extract from *Barringtonia acutangula* inhibits the 11β-hydroxysteroid dehydrogenase type 1 (11β-HSD1); however, this enzyme has been shown to decrease the expression of PPARγ and increase that of PPARα [76].

The leaf extract from *Melissa officinalis* reduces the mRNA levels of genes involved in lipogenesis, such as the fatty acid synthase (FAS), stearoyl-CoA desaturase 1 (SCD1), and sterol regulatory element-binding protein 1c (SREBP-1c) [77]. The extract of *M. officinalis* acts via the C/EBPα and the PPAR pathway [78]. The extract ALS-L1023 confirmed that the effects of nutraceuticals present in this herb reduce weight gain. This extract, which increases the level of AMPKα2, alleviates abdominal obesity and insulin resistance in C57BL/6J mice by increasing the mRNA levels of carnitine palmitoyltransferase 1 (CPT-1), medium-chain acyl-coenzyme A dehydrogenase (MCAD) [79]. Thus, the effects of the herbs depend on the different nutraceuticals obtained according to solvents used to prepare the extracts.

An extract from *Momordica charantia* seed exhibits lipid-lowering activity by reducing the insulin-stimulated IRS-1 tyrosine phosphorylation; however, IRS-1 is linked to PPARγ subunit [80]. In particular the IRS-1 variant (C189T > G) was associated with post-weaning body weight and body weight gains [81] (Table 5).

Overall, these results confirm that many mechanisms regulate, directly or indirectly, the cellular fat mass. Moreover, these mechanisms interfere each other, confirming the difficulties in treating obesity over a long term due to a complex pattern of inter and intra-cellular signals (Figure 2).

### 2.3. Reducing Inflammation

Chronic inflammation impacts the fat storage in adipose tissue, resulting in free fatty acid and triglyceride excesses in bloodstream and induction of insulin resistance in muscle and liver. Adipocytes crosstalk with macrophages, which are able to infiltrate fat masses, leading to obesity, inducing ectopic fat deposition in these tissues. [82]. Enlarged adipocytes, together with the infiltrated macrophages, act in a synergistic manner to cause aberrant production of pro-inflammatory molecules including inducible nitric oxide, cytokines such as TNF-α, interleukine-6 (IL-6) and the chemokine monocyte chemoattractant protein-1 (MCP-1). Both TNF-α and IL-6 can lead to insulin resistance by triggering different key steps in the insulin signalling pathway. However, several different factors can induce inflammation, especially chronic low-grade inflammation, and cause obesity. Proinflammatory pathways, free radical increase, and epigenetic modulation of some genes, such as methylation of the IGF-1 encoding gene, can affect the growth of fat masses. These pathways involve different protein complexes and metabolites, for example by activation of inflammasome, kinases, and related coactivators.

#### 2.3.1. Activation of Inflammosome

The inflammasome complex is a multimer consisting of pattern-recognition receptors (PRR); each one is a monomer sensor that responds to inflammatory stimuli; after stimulation, they oligomerize and form a pro-caspase-1 activation platform. Several members of PRRs have been confirmed to form inflammasomes; among these are the NLRP3, a member of the leucine-rich repeat (LRR)-containing proteins (NLRP) [83]. The NLRP3 inflammasome is activated by diverse stimuli and multiple molecular and cellular events, including ionic flux, mitochondrial dysfunction, such as the production of reactive oxygen species (ROS), and lysosomal damage. In obese individuals, stressed adipocyte and macrophage cells respond by activating the NLRP3 inflammasome, which modulates the MAPK signalling or decreases the expression of proinflammatory cytokines such as TNF-α and IL-6, and could improve inflammation-related diseases. However, aberrant activation of NLRP3 inflammasome has been related to obesity. An extract from *Coptidis rhizoma* has been shown to have successful effects in obese rats affected by glomerulopathy [84]. Subsequent studies on Coptidis rhizome, used in oriental medicine, showed that its anti-inflammatory activity is linked to berberine derivatives [85]. The analyses of its biological effects revealed that it inhibits dysregulated NLRP3 inflammasome complex. *Morus alba* is traditionally used as an antioxidant and anti-inflammatory agent, and it shows an anti-obesity effect [86,87]. Some research describes the effects of these herbs on TNF-α and IL-6. Recently, administration to HFD-induced obese C57BL/6 mice of an extract of leaves from *M. alba,* fermented with fungus *Cordyceps militaris*, has been shown to prevent fat accumulation through the inhibition of lipogenesis and stimulation of lipolysis [88]. However, the fermented powder can contain by-products from fungal fermentation, involved in the mechanism of action, whose secondary effects are still unknown (Table 6).

#### 2.3.2. TNF-α Pathway

The activation of immune cells, such as macrophages, is stimulated by secretion of TNF-α which, in paracrine and autocrine manner, regulates a number of critical cell functions including cell proliferation, survival, differentiation, and apoptosis. Macrophages are the major producers of TNF-α and interestingly are also highly responsive to TNF-α itself. Aberrant TNF-α production and TNF receptor signaling have been associated with the pathogenesis of several diseases, including rheumatoid arthritis, Crohn’s disease, atherosclerosis, psoriasis, sepsis, diabetes, and obesity. 

*Curcuma longa* rhizome contains the curcuminoids, which reduce inflammation and improve obesity. Curcuminoids have also been proposed as browning promoting agents, leading to weight loss by suppressing chronic inflammation in white adipocytes through the modulation of TNF-α [89]. Unfortunately, curcuminoids are effective at high dosages which induce undesirable effects, and therefore some Food Safety Agencies have decreased the amounts allowed in the formulation of food supplements.

*Nelumbo nucifera* contains quercetin-3-O-ß-glucuronide (Q3GA) and quercetin; experiments conducted in vitro confirmed that quercetin and Q3GA affect lipid metabolism by promoting triglyceride degradation through inhibition of the cAMP pathway which in turn blocks the TNF-α pathway [90].

*Garcinia mangostana* contains xanthones, which act on inflammatory and anti-inflammatory pathways [91].

The total fraction of soluble compounds present in the aqueous extract of the *Cichorium intybus* was added to the diet of obese diabetic mice, reducing obesity [92] (Table 7).

#### 2.3.3. MAPK Signalling

MAPK pathways are signaling modules that transduce extracellular and intracellular signals to regulatory networks within the cell via the phosphorylation of key protein targets. Each cascade is activated by extracellular signals which lead to the activation of the MAPK, which in turn become substrates of a second specific kinase (MAPKKK). A subsequent third kinase (MAPKK) completes the signal transduction. However, three different families of kinases, ERKs, JNKs, and p38/SAPKs, respond to different extracellular signals of stress [93]. Although they show some differences in the modulation of different genes, all these pathways can induce obesity. We report that some herbal extracts, having the MAPK pathways as target, are able to reduce obesity.

A *Cordyceps militaris* extract, containing cordycepin, has been shown to prevent overweight, fat accumulation, liver hypertrophy, and lowered triglyceride levels by modulating gene expression. The targeted genes were related to the insulin signalling pathway, insulin resistance, the MARK signalling pathway and the PI3K–Akt signalling pathway [94].

*Sorghum bicolor* has been found to contain a fraction extract rich in 3-Deoxyanthocyanines; the extract reduces inflammation and the ROS cascade and promotes the gut *Bacteroidetes* which are highly present in lean individuals [95]. However, *S. bicolor* has been shown to activate the MAPK cascade, suggesting an intracellular mechanism to reduce lipid accumulation [96] (Table 8).

#### 2.3.4. Production Control and Scavenging of Free Radicals

High free radical levels arise after eating and they increase the sense of appetite, so that antioxidants could be useful to reduce obesity. There is evidence that excessive ROS contribute to develop metabolic disorders leading to inflammation and obesity [97]. Antioxidant natural extracts could ameliorate inflammation, especially low-chronic grade one, and reduce appetite by controlling the level of ROS and their activity [98]. However, in spite of a large number of natural antioxidant extracts, only few of them seem to be effective to treat obesity. A natural ent-Kaurane diterpenoid, Eriocalyxin B, extracted from *Isodon eriocalyx*, showing anti-inflammatory activity, was found inhibits adipogenesis in 3T3-L1 adipocytes [99]. Moreover, the role played by nitric oxide (NO) reactive species in obesity, received similar attention. A raw non-polar extract from the leaves of *Angelica keiskei* improves obesity by reducing the amount of NO [100,101]. NO has proposed as gaseous signalling molecule linked to onset of obesity by inducing low-chronic grade inflammation [102,103]. However, an extract rich in extract chalcones also decreases the TNF-α levels and in turn reduces insulin resistance [104,105] (Table 9).

### 2.4. Regulation of Lipase Activity

Fatty acids are hydrolysed at gastrointestinal level and transported to liver where they are re-esterified into triglycerides and transported to other tissue by lipoproteins. Pancreatic lipase and Lipoprotein lipase are responsible for the conversion of fats into monoglyceride and free fatty acids, which can across the enterocytes. The fat can reach the adipocytes, where it is accumulated causing the enlargement of these cells. Thus, the inhibition of lipases and related enzymes is another strategy to reduce adipose tissue hyperplasia [106]. Delaying fat adsorption or inhibiting fat digestion, by suppressing lipase activity, decrease obesity as shown by the Orlistat, but unfortunately this medicine has secondary undesired effects [106]. The research of lipase inhibitors has been investigated in 76 plant species resulting able to inhibit lipases. Among them we report *Hieracium and Pilosella*, which are two strictly related species diffused as popular remedies of obesity [107].

*Taraxacum officinale*, traditionally known as Dandelion, contains several active compounds such as taraxasterol, chlorogenic acid, α-amyrin, which exhibit anti-inflammatory effects, inhibiting NO synthase and COX-2 protein expression. Often, some nutraceuticals depending on extract fraction, in a dose dependent manner, inhibit pancreatic lipase and in turn improve fat deposition [108].

Recently, an extract from *Ginkgo biloba* leaf was described to contain a novel pancreatic lipase inhibitor, but this is the only report describing its anti-lipolytic activity [109].

Moreover, ginkgolides are considered the most interesting compounds improving insulin resistance via Toll-like receptors signalling cascades [110].

*Prunus armeniaca* leaves contain phenolic compounds that show several activities, all of them have positive effect on reducing obesity, in particular inhibiting the pancreatic lipase [111].

*Rhus verniciflua* extracts reduce appetite. An extract from this plant was found to act on dopaminergic cells, to inhibit the activity of glycolytic enzymes, such as alpha-glucosidase, and consequently to suppress obesity in an animal model [112]. A leaf extract from *Rhus verniciflua*, rich in quercetin derivatives, was found lowering SREBP-1 and triglyceride levels. It has been observed that PPARs and SREBP-1 interaction reduces lipid deposition [113]. The extract suppresses obesity in high fat diet-induced obese mice [114]. *Rhus verniciflua* ethanol extract revealed a strong alpha-glucosidase inhibitory activity, which was able to reduce weight gain and improve insulin resistance [115]. Furthermore, a *R. verniciflua* Stockes extract, used as traditional medicine in Korea, exhibited lipid-lowering effects by lowering SREBP-1 and triglyceride levels, and promoting the activation of PPARs and AMPK in an in vitro model of non-alcoholic fatty liver disease [116].

An extract from *Morus nigra* acts on hormone-sensitive lipase showing anti-obesity effects [87,117] (Table 10).

### 2.5. Effectors on Neuromodulators

Adipose tissue is also an active endocrine organ, diffused through body tissues, that regulate systemic metabolism via crosstalk with multiple peripheral tissues and the central nervous system [118]. The adipokines are the hormones produced by adipocytes, and their interactions with neuromodulators is one of the main targets of anti-obesity treatments. Dysregulation of peripheral and central hormones leads to an imbalance of energy expenditure. Oxidative stress in hypothalamus is one of the causes of an unnatural request for food. The increased level of ROS and NO in the first-order neurons into arcuate nucleus, and the aberrant functioning of orexigenic Agouti-related protein (AgRP)/Neuropeptide Y (NPY), anorexigenic pro-opiomelanocortin (POMC)/cocaine- and amphetamine-regulated transcript (CART) can lead to obesity. The neurosignals regulate the demand for food intake and energy expenditure [119]. Among successful species, *Panax ginseng* is traditionally used for its ability to induce the body to adapt to different stresses, and its neuromodulatory activity is considered useful in treatment of obesity. The assumption of campesterol, β-elemene, ginsenoside Rb1, contained in *P. ginseng* have been related to modulation of neuropeptide genes that regulate energy balance and food intake. The analysis of data suggests that *P.gineng* can regulate neuropeptides associated with appetite [120,121].

Another study on *Paullinia cupana* showed its ability to suppresse appetite by stimulating the central nervous system through the action of methylxanthine [122]. Xanthines stimulate lipolysis and inhibit adipogenesis and, more generally, as those present in high amounts in teas containing cathechin and caffeine, have been reported to support body weight control [123]. The xanthines can function by two different but convergent mechanisms, by increasing the ghrelin or decreasing the TNF-α levels, respectively [124] (Table 11).

## 3. Conclusions

The accumulation of fat mass is caused by multiple genetic, epigenetic and lifestyle factors. Obesity correlates with hyperplasia and hypertrophy of white adipose tissue, which can be linked to hyperlipidemia, increased lipogenesis, chronic inflammation, lack of exercise, and a high-fat or simple carbohydrate-rich diet. The regulation of each of these conditions occurs both at the intracellular and intercellular level. Several intracellular pathways are involved in the activation of adipogenesis and lipolysis, such as those of AMPK and PPARs. Other pathways, such as those of MAPKs and PI3K/AKT, are indirectly associated with obesity by activating specific genes that regulate cell differentiation or the inflammatory response.

Moreover, the intercellular regulation of the adipocyte metabolism acts through endocrine adipokines and neurohormones, regulating cell crosstalk and affecting eating behaviours. The search for key pathways and useful substances able to control all these mechanisms that could be called the obesosome is challenging. It has been seen that the activation of AMPKα is useful to fight obesity; however, at the same time, AMPKα inhibition protects against arterial thrombosis. On the other hand, the activation of PPARα reduces weight gain, whereas the inhibition of PPARγ decreases lipogenesis; unfortunately, current anti-obesity agents are not selective effectors for each subunit. Furthermore, this non-specific inhibition of PPARs can induce inflammation, causing weight gain and obesity. Nutraceuticals blocking inflammatory pathways have been shown to increase insulin resistance and activate the polarization of macrophages, which infiltrate adipose tissue and induce an increase of fat mass. This scenario suggests that there is a need for multiple targeted anti-obesity agents. It is a hard challenge to find a single drug having all functional requirements. A solution to act on obesosome could come from nutraceutical compositions. Certainly, to find a mixture of herbal nutraceuticals, with synergistic and selective activity, will require a new technological approach to the problem.

## Figures and Tables

**Figure 1 nutrients-14-05099-f001:**
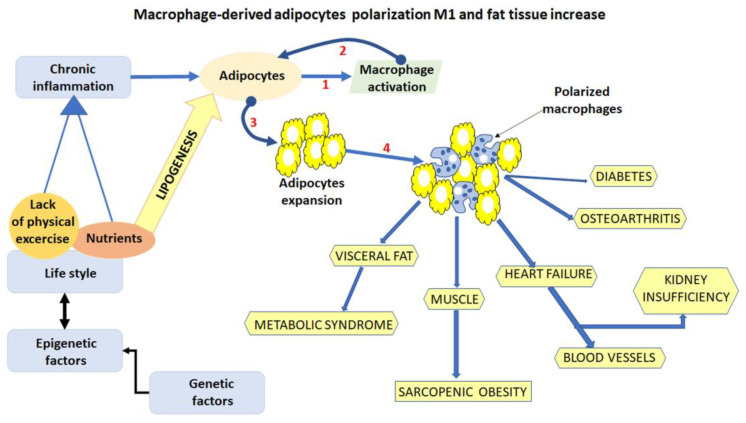
Schematic representation of the fat mass expansion and related diseases secondary to obesity. Adipocyte crosstalk macrophage (1) which in turn stimulate adipocyte inflammation (2) and expansion (3); inflamed hypertrophic adipocytes promote the accumulation of M1 polarized macrophages in adipose tissue (4).

**Figure 2 nutrients-14-05099-f002:**
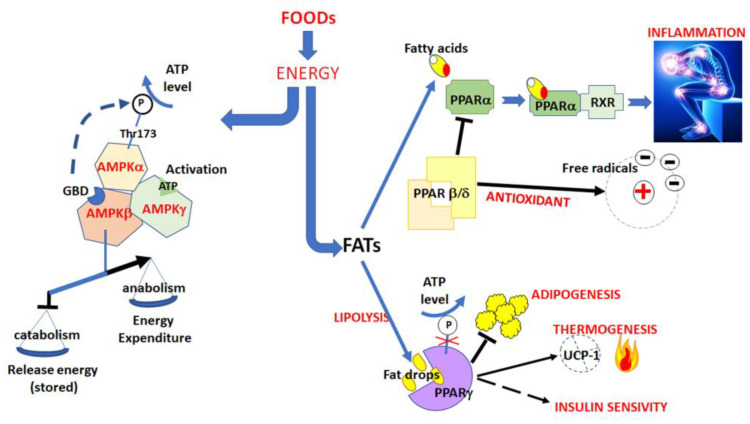
Effects of nutrition on AMPK and PPARs pathways.

**Table 1 nutrients-14-05099-t001:** Herbs containing nutraceuticals acting as AMPK effectors.

Source	Proposed Mechanism	Target	Effects
*Gelidium elegans* [20,21]	Unknown	AMPK-PRDM16	Anti-obesity
*Iris rossii Baker* [36,37]	Inhibits pre-adipocytes differentiation	AMPK	Decreases lipid accumulation
*Panax ginseng* [25,26]	Increases AMPK phosphorylation	AMPK PRDM16 PGC1α	Improves insulin sensitivity and glucose metabolism
*Paullinia cupana* [22,18]	Stimulates thermogenesis	AMPK PPARγ coactivator 1-alpha (PGC1-α) UCP-1	Anti-hyperlipidemic
*Pueraria montana* var *lobata* [29,30]	Regulation of brown fat activity	PPAR pathway UCP1	Stimulates thermogenesis
*Salvia hispanica* L. [38,39]	Increases protein kinase B 1 (AKT1)[pS473]	AMPK mRNA HSP, PGC-1α Forkhead box protein 1 (FOXO1)	Improves glucose and insulin tolerance
*Scutellaria baicalensis* [34,35]	Hemostasis of glucose and lipid metabolisms	(WNT)/β-catenin pathway AMPK PPARγ	Anti-hyperlipidemic
*Zingiber officinalis* [32,33]	Upregulating β-oxidation	AMPK	Controls fat accumulation

**Table 2 nutrients-14-05099-t002:** Herbs containing nutraceuticals which indirectly activate AMPK.

Source	Proposed Mechanism	Target	Effects
*Angelicae sinensis* [44,45]	Controls adipocytes expansion	FTO gene AMPK	Suppresses body weight gain
*Solanum melongena* or *aethiopicum* [46,47]	Controls the adipocytes in hypothalamus	FTO gene AMPK	Decreases feed consumption

**Table 3 nutrients-14-05099-t003:** Herbs containing nutraceuticals acting on PPAR pathway.

Source	Proposed Mechanism	Target	Effects
*Astragali membranaceus* var. *mongholicus* [64,65]	Increases thermogenesis	PPARγ	Antidiabetic/anti-inflammatory
*Citrus reticulata* [60,61]	Increases hepatic fatty acid oxidation	PPARα	Anti-hyperlipidemic activity
*Nigella sativa* [62,63]	Reduces adipogenesis	PPARγ TNF-α Adipokines	Decreases appetite Reduces body weight
*Rosa rugosa* [58,59]	AMPK pathway activation	PPARα agonist	Control of dyslipidemia
*Spermacoce hispida* [57]	Decreases lipid accumulation	PPARα	Anti-hyperlipidemic activity

**Table 4 nutrients-14-05099-t004:** Herbs containing nutraceuticals acting on C/EBP pathway.

Source	Proposed Mechanism	Target	Effects
*Angelica dahuricae* [70,71]	Inhibits adipogenesis	C/EBPβ signaling	Controls adipose tissue
*Bidens pilosa* [72]	Decreases the adipogenesis and lipid accumulation	C/EBPs PPARγ Egr2	Decreases fat content
*Theobroma cacao* [68,69]	Stimulates thermogenesis	C/EBPα	Controls body weight
*Coptis chinensis* [73]	Inhibits adipocyte differentiation	C/EBPα PPARγ	Reduces obesity
*Isodon adenantha* [74]	Inhibits adipogenesis decreasing the ROS amount	C/EBPβ signaling	Controls adipose tissue
*Ziziphus jujuba* [67]	Affects adipogenic differentiation	C/EBP PPARγ PI3K/AKT	Anti-adipogenic

**Table 5 nutrients-14-05099-t005:** Herbs containing nutraceuticals indirectly acting on PPAR pathway.

Source	Proposed Mechanism	Target	Effects
*Barringtonia acutangular* [76]	Controls metabolism of lipids	11β-HSD1	Reduces hyperlipidemia
*Melissa officinalis* [77,78]	Increases fatty acid β-oxidation and decreases lipogenesis	FAS SREBP-1 CPT-1	Reduces visceral adiposity
*Momordica charantia* [81]	Insulin signalling	IRS-1	Reduces weight gain

**Table 6 nutrients-14-05099-t006:** Herbal extracts active on NLRP3 inflammasome.

Source	Proposed Mechanism	Target	Effects
*Coptidis rhizoma* [84,85]	Decreases adipose tissue macrophages	NLRP3	Reduces obesity
*Morus alba* [86,87]	Adipocyte differentiation	NLRP3 PPARγ and C/EBPα	Reduces dyslipidemia

**Table 7 nutrients-14-05099-t007:** Herbal extracts acting on TNF-α pathway.

Source	Proposed Mechanism	Target	Effects
*Curcuma longa* [89]	Anti-inflammatory	TNF-α	Reduces obesity
*Garcinia mangostana* [91]	Anti-inflammatory	TNF-α IL-6	Reduces body weight
*Nelumbo nucifera* [90]	Inhibition of cAMP pathway	TNF-α, leptin, insulin	Prevents triglyceride accumulation and promote lipolysis
*Cichorium intybus* [92]	GDF-15 signalling pathways	PI3K/AKT	Weight reduction

**Table 8 nutrients-14-05099-t008:** Herbal extracts acting on MAPK signaling.

Source	Proposed Mechanism	Target	Effect
*Cordyceps militaris* [94]	Decreases population of *negativebacillus*	MAPK signaling pathway; PI3K–Akt signaling pathway	Reduce body weight, fat accumulation. Stimulate lipolysis
*Sorghum bicolor* [95,96]	Inhibits preadipocyte differentiation	MAPK signaling production of ROS	Reduce intracellular lipid accumulation

**Table 9 nutrients-14-05099-t009:** Herbal extracts reducing oxidative stress.

Source	Proposed Mechanism	Target	Effect
*Angelica keiskei* [100,101]	Anti-inflammatory	TNF-α	Reduce gains in body weight
*Isodon eriocalyx* [99]	Anti-inflammatory	NF-κB	Inhibit adipogenesis

**Table 10 nutrients-14-05099-t010:** Herbal extracts inhibiting lipase activity.

Source	Proposed Mechanism	Target	Effects
*Ginkgo biloba* [109,110]	Adipocyte hypertrophy	FAS Perilipin 1	Reduce the adipocyte volume
*Hieracium* sp. [107]	Decrease lypolysis	Pancreatic lipase	Decrease fat accumulation
*Morus nigra* [87,117]	Leptin-stimulated lipolysis	Hormone-sensitive lipase FAS	Decrease fat mass
*Prunus armeniaca* [111]	Inhibit adipogenesis	pancreatic lipase	Prevention of obesity
*Rhus verniciflua* [112,113,114,115,116]	Inhibit nonesterified fatty acid and glycerol absorption	SREBP1 alpha-glucosidase	Reduce body weight gain
*Taraxacum officinale* [108]	Increase plasma superoxide radical scavenging	Alkaline phosphatase	Decrease in lipid peroxidation

**Table 11 nutrients-14-05099-t011:** Herbal extracts with neuroactivity.

Source	Proposed Mechanism	Target	Effects
*Panax ginseng* [120,121]	Regulates appetite-related neuropeptides	Pro-opiomelanocortin, cholecystokinin, and cocaine- and agouti-related protein, neuropeptide Y	Decreases adipogenesis
*Paullinia cupana* [122]	Inhibits adipogenesis	Ghrelin	Decreases fat mass

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
