# Peer review of "Nutraceuticals and the Network of Obesity Modulators"

_nutrients, 2022, doi:10.3390/nu14235099_

Round 1
Reviewer 1 Report
Dear
This review contains Nutraceuticals as obesity modulators.
However, this review has several deficiencies as follows and needs a major revision.
1. Many plant sources are listed in the Tables, but not in the main text. Please, match the table with the main text.
2. Please, add a few references (not only one) for each plant in the Tables.
3. Please, enlarge the figure images and clarify words.
Author Response
REVIEW 1
This review contains Nutraceuticals as obesity modulators.
However, this review has several deficiencies as follows and needs a major revision.
- Many plant sources are listed in the Tables, but not in the main text. Please, match the table with the main text.
Thank you very much for your comment, we matched the plant reported in the main text with those reported in tables
- Please, add a few references (not only one) for each plant in the Tables.
Where possible, we added more references for each plant in the tables. In some cases we have found only one reference for a specific plant acting on a specific pathway, for those plant we added only one reference.
- Please, enlarge the figure images and clarify words.
We enlarged both the figures and the size of font. The words are now much more readable
We would like to thank the Reviewer for the accurate revision of our manuscript and hope that the revised version is suitable for publication.
Reviewer 2 Report
It is a very interesting and up-to see-date review which presents a different pathway that are included in adipogenesis and lipolysis.
It is a comprehensive review which fits well to the scope of Nutrients journal.
Tables and figures are clear and informative.
Author Response
REVIEW 2
It is a very interesting and up-to see-date review which presents a different pathway that are included in adipogenesis and lipolysis.
It is a comprehensive review which fits well to the scope of Nutrients journal.
Tables and figures are clear and informative.
We would like to thank the Reviewer for the accurate revision of our manuscript and for her/his very positive comments.